# Hydrolysis of Beef Sarcoplasmic Protein by Dry-Aged Beef-Isolated *Penicillium oxalicum* and Its Associated Metabolic Pathways

**DOI:** 10.3390/foods13071038

**Published:** 2024-03-28

**Authors:** Yujia Liu, Depeng Sun, Anqi Peng, Tingyu Li, Hongmei Li, Baide Mu, Juan Wang, Mingxun Cui, Chunxiang Piao, Guanhao Li

**Affiliations:** 1Key Innovation Laboratory for Deep and Intensive Processing of Yanbian High Quality Beef, Ministry of Agriculture and Rural Affairs, Yanji 133002, China; 2022010677@ybu.edu.cn (Y.L.); 2022001108@ybu.edu.cn (D.S.); penganqi@intco.com (A.P.); 0000008821@ybu.edu.cn (T.L.); hmli2018@ybu.edu.cn (H.L.); mubaide@ybu.edu.cn (B.M.); wangjuan@ybu.edu.cn (J.W.); 0000001384@ybu.edu.cn (M.C.); 2College of Agriculture, Yanbian University, Yanji 133002, China

**Keywords:** Yanbian cattle, dry aging, *Penicillium oxalicum*, sarcoplasmic protein

## Abstract

Yanbian cattle have a unique meat flavor, and high-grade meat is in short supply. Therefore, in this study, we aimed to improve the added value of Yanbian cattle low-fat meat and provide a theoretical reference for the subsequent development of an excellent starter. Rump meat from Yanbian cattle was dry-aged and then screened for protease-producing fungi. Three protease-producing fungi (*Yarrowia hollandica* (D4 and D11), *Penicillium oxalicum* (D5), and *Meesziomyces ophidis* (D20)) were isolated from 40 d dry-aged beef samples, and their ability to hydrolyze proteins was determined using bovine sarcoplasmic protein extract. SDS-PAGE showed that the ability of *Penicillium oxalicum* (D5) to degrade proteins was stronger than the other two fungi. In addition, the volatile component content of sarcoplasmic proteins in the D5 group was the highest (45.47%) and comprised the most species (26 types). Metabolic pathway analysis of the fermentation broth showed that phenylalanine, tyrosine, and tryptophan biosynthesis was the most closely related metabolic pathway in sarcoplasmic protein fermentation by *Penicillium oxalicum* (D5). Dry-aged beef-isolated *Penicillium oxalicum* serves as a potential starter culture for the fermentation of meat products.

## 1. Introduction

Beef is a rich source of essential proteins, fats, and amino acids, and its consumption is gradually increasing [1,2]. The aging process of beef affects its edible qualities, such as tenderness, juiciness, and flavor. Long-term meat aging significantly improves the meat quality, particularly tenderness, via disruption of the muscle structure by endogenous proteolytic systems [3]. Dry and wet aging are two commonly used meat aging methods. Compared with wet aging, dry aging improves the flavor and taste of meat considerably [4,5,6,7]. The formation of free amino acids and flavor during dry aging is mainly caused by microbial fermentation [8]. Currently, some of these strains with proteolytic and lipolytic abilities can be selected according to the production requirements [9], thus improving the hydrolysis of proteins and fats in beef during dry aging. For example, Ryu et al. [10] suggested that yeast and mold play important roles in the taste and flavor quality of dry-aged beef. Molds and yeasts release proteases that penetrate the meat, thereby improving its flavor and tenderness [11] and increasing the content of free amino acids and fatty acids [12]. It can also shorten the production cycle and reduce losses; therefore, natural fermentation has been gradually replaced by artificial inoculation [13,14]. Many commercial starters are available, for example, *Lactobacillus sakei* and *Staphylococcus xylosus* [15]. However, commercial meat starter cultures are not suitable for all fermentation conditions and cannot compete with local microbial populations, limiting their application in meat product processing [16]. Dashdorj et al. [8] found a mold called *Thamnidium* in dry-aging beef, which produces proteases to degrade muscle proteins and improve tenderness and taste. Oh et al. [17] isolated molds and yeasts from dry-aged beef that exhibit strong protein and fat hydrolysis activities and promote the decomposition of myofibrils. Therefore, it is important to select suitable starter cultures from the dominant local microbial communities, as they are well adapted to the meat environment and offer metabolic advantages [18].

However, in China, only a few studies have evaluated the development and application of dry-aged fermentation inoculants. Therefore, in this study, we aimed to determine whether beneficial microorganisms with protease-producing activity, isolated from dry aged-beef, could be used to add value to Yanbian cattle low-fat meat.

## 2. Materials and Methods

### 2.1. Processing and Sampling of Dry-Aged Beef

Three pieces of rump meat (weight 1.5 ± 0.2 kg) from three Yanbian cattle with similar characteristics (male, 16–20 months) were selected for this study (Benfu Halal Meat Co., Ltd., Yanji, China). Immediately after slaughtering, according to the slaughtering method of Benfu Halal Meat Co., Ltd., Yanji, China, the animals were moved to the fermentation room at 2–4 °C. One day before experimentation, the fermentation room was fully disinfected using alcohol and sterilized using UV light irradiation. Dry aging of the obtained rump meat was carried out for 40 d, as previously described (0–10 d: 2 ± 1 °C, 75 ± 5% RH; 11–20 d: 4 ± 1 °C, 75 ± 5% RH; 21–40 d: 4 ± 1 °C, 85 ± 5% RH), at a wind speed of 1.2 m/s [19]. Samples were collected after 40 d of dry aging. An amount of 10 g of meat pieces, 1 cm from the crust of the sample, was placed in a sterile homogeneous bag in a sterile environment. The remaining meat samples were also placed in sterile homogeneous bags at 2–4 °C and stored at −80 °C until further use.

### 2.2. Screening of Protease-Producing Fungi

#### 2.2.1. Isolation and Preliminary Screening of Protease-Producing Fungi

We screened for protease-producing fungi in the dry-aged 40 d beef samples. In a sterile environment, 10 g of meat pieces, 1 cm from the crust of the sample, were placed in a sterile homogeneous bag (Bkmam Biotechnology Co., Ltd., Changde, China), and nine times the volume of 0.85% sterile saline (Bkmam Biotechnology Co., Ltd., Changde, China) was added and homogenized for 5 min on a beating homogenizer BagMixer400 (Interscience, Paris, France). Subsequently, the homogenized sample was diluted with 0.85% sterile physiological salt gradient and plated on Bengal red medium. The plated samples were cultured at 30 °C for 3–7 d. All single colonies were inoculated on potato solid medium, containing 1% skim milk, and cultured at 30 °C for 3–7 d. The isolated strains were screened for the presence or absence of a proteolytic cycle. In this study, 1% milk medium was used to isolate high-yield protease-producing fungi from dry-aged beef. The colonies with hydrolysis circles were selected from 1% milk medium for isolation and purification, and 27 strains were obtained, numbered D1–D27. After purification, the isolated strains were re-cultured in potato solid medium containing 1% milk, and 10 strains with larger hydrolysis circles were selected for subsequent experiments [20].

#### 2.2.2. Rescreening and Identification of Protease-Producing Fungi

The determination was carried out using the Folin phenol method [21]. The standard curve was drawn with the absorbance (A) as the ordinate and the 100 μg/mL L-tyrosine standard stock solution as the abscissa. The strain was cultured in a potato liquid medium at 30 °C for 48 h, and the resulting culture was centrifuged at 10,000× *g* for 10 min. The supernatant was collected and used for enzyme screening, as previously described [22]. The absorbance of all samples was measured at 680 nm using a spectrophotometer, and the tyrosine concentration was extrapolated using a standard curve. One enzyme activity unit U was defined as the amount of enzyme required to produce, via casein hydrolysis, 1 μg of tyrosine per minute. The formula used is as follows:$$U=\frac{A\times 4\times N}{10}$$

*A*: The absorbance of the sample; tyrosine value obtained using the tyrosine standard curve (μg).

4: Reaction solution (mL).

10: Reaction time (min).

*N*: Dilution fold of enzyme solution.

The Internal Transcribed Spacer (ITS) amplification products of the strains were identified by Jilin Kumei Biotechnology Co., Ltd.(Jilin, China) in China via DNA sequencing. The obtained sequence reads were analyzed via BLAST homology comparison using the GenBank database and the NCBI website (www.NCBI.nlm.nih.gov), accessed on 5 October 2022. Finally, a phylogenetic tree was constructed to determine the phylogenetic relationship between the identified species using MEGA 6.0 software.

### 2.3. Preparation of Sarcoplasmic Protein Extracts and Cell Suspensions

Sarcoplasmic proteins were extracted as previously described [23], with slight modifications. Briefly, 1 g of fresh rump beef was added to 10 mL of 50 mM Tris-HCl buffer (pH = 8.0), homogenized in an ice bath for 2 min, centrifuged at 10,000× *g* at 4 °C for 20 min, and then the supernatant was filtered through gauze into sterile centrifuge tubes for use as a sarcoplasmic protein extract.

The supernatant was supplemented with 2% NaCl (7647-14-5, National Pharmaceutical Group Chemical Reagent Co., Ltd., Shanghai, China) and 1% glucose (50-99-7, National Pharmaceutical Group Chemical Reagent Co., Ltd., Shanghai, China), and then passed through a 0.22 μm sterile filter membrane. After the potato plate was tested for sterility, the samples were cultured in test tubes containing 10 mL of potato liquid culture medium (three tubes per sample). The concentration of each strain suspension was 10^7^ CFU/mL. For the control sample, the culture medium was not inoculated. The cells were cultured at 30 °C for 4 d. The fermentation performance of strains with high protein hydrolysis activity was evaluated by the in vitro hydrolysis of sarcoplasmic proteins. The strains with high protein hydrolysis activity were applied to sarcoplasmic proteins to study their ability to degrade sarcoplasmic proteins and to utilize amino acids in sarcoplasmic proteins to generate volatile flavor components. Inoculation of 2% of single strains of *Yarrowia hollandica* D4 and D11, *Penicillium oxalicum* D5, and *Meesziomyces ophidis* D20 was recorded as groups D4, D5, D11, and D20, respectively. A mixed strain (D4:D5:D11:D20) in a 1:1:1:1 ratio, recorded as the F group, inoculated at 2%, was also tested. The composition and content of fermentation broth SDS-PAGE, free amino acids, and volatile flavor components were determined.

### 2.4. Gel Electrophoresis

To analyze the degree of sarcoplasmic protein degradation, SDS-PAGE was performed as previously described [24] using the Mini-PROTEAN electrophoresis system (Bio-Rad Co., Ltd., Hercules, CA, USA) and the rainbow 245 broad-spectrum protein marker. A total of 60 μL of each extraction solution was mixed with 20 μL of 4× protein loading buffer (Solarbio Science & Technology, Beijing, China) and heated at 95 °C for 5 min. The separation gel concentration was 12%; the concentrated gel concentration was 5%; and the sample volume was 10 μL. After electrophoresis, the gel was immersed in a 0.25% Coomassie Brilliant Blue R250 staining solution for 2 h and then destained in a decolorization solution (methanol/acetic acid/water = 5:4:1) until the background was clear. After decolorization, a gel analyzer was used to capture photographs.

### 2.5. Free Amino Acids and Volatile Flavor Compounds

#### 2.5.1. Determination of Free Amino Acids

The sarcoplasmic protein fermentation broth from each group fermented for 4 d was selected, and 10 mL of the fermentation broth was added to 20 mL of 4% trichloroacetic acid (TCA) and incubated at 37 °C for 30 min. Next, the supernatant was filtered through a 0.45 μm organic filtration membrane (Jinteng Experimental Equipment Co., Ltd., Tianjin, China). The amino acid content was determined using an automatic amino acid analyzer (Hitachi Co., Ltd., Tokyo, Japan), and the results were expressed as mg/100 g of the sample.

#### 2.5.2. Extraction of Volatile Flavor Components

The sarcoplasmic protein fermentation broth from each group fermented for 4 d was selected, and 2 mL of the fermentation broth was added to a 15 mL headspace bottle and placed in an equilibrium gas at room temperature for 15 min. With a 50 μm solid-phase microextraction fiber head (PDMS/DVB, Supeclo, PA, USA), the volatile flavor components were extracted for 30 min in a sand bath at 60 °C. GC (QP2010 Plus, Shimadzu Co., Ltd., Kyoto, Japan) conditions were as follows: DB-5MS capillary column (Agilent Technologies, Santa Clara, CA, USA) with an inner diameter of 0.25 mm, a length of 30.0 m, and a film thickness of 0.25 μm; the temperature of the injection port was 250 °C; the sample was injected without shunt; the carrier gas was helium; the purge flow rate was 3.0 mL/min; and the flow rate of the column was 1 mL/min. Temperature rise program (48 min in total): initial temperature 40 °C held for 10 min, increased to 200 °C at 5 °C/min, increased to 280 °C at 20 °C/min and held for 5 min, MS conditions: interface temperature 280 °C, ion source temperature 200 °C, solvent delay time of 4 min, electron energy of 70 eV, and scanning mass range of *m/z* 40–550. The compounds were qualitatively analyzed by searching the NIST10 and NIST10s libraries. Compounds with matching degrees greater than 75 were selected, and the relative content of each compound was calculated using the peak area normalization method [25,26].

### 2.6. Metabolic Pathway Analysis of Penicillium oxalicum (D5) Fermentation Broth

We conducted a metabolic pathway analysis of fermentation broth by selecting strains with more pronounced effects on protein and flavor. We explored the correlation between *Penicillium oxalicum* and key metabolites to elucidate the contribution of *Penicillium oxalicum* to the unique flavor of dry-aged beef and the metabolic pathways that influence the flavor of dry-aged beef. Orthogonal partial least squares discriminant analysis was performed by selecting metabolite names and contents as variables using SIMCA 14.0 software to visualize free amino acid and volatile flavor component data. Based on the PLS-DA model and one-way analysis of variance (ANOVA), the projected variable importance values and P values were calculated, and significantly different metabolites were identified based on the principles of VIP > 1 and *p* < 0.05. Finally, the names of significantly different metabolites were entered into the Metabo Analyst 4.0 website (http://metaboanalyst.ca, accessed on 18 March 2023) for enrichment analysis, and the metabolic pathways were combined with the Kyoto Encyclopedia of Genes and Genomes (KEGG) database for metabolic pathway analysis.

### 2.7. Statistical Analysis

All experiments were carried out with three replicates. IBM SPSS Statistics 23 (SPSS, Chicago, IL, USA) was used for variance analysis, and the Tukey test was used to determine the significant differences (*p* < 0.05), and the results were expressed as mean ± standard deviation. A phylogenetic tree was constructed using MEGA 6.0. GraphPad Prism software (version 8.0) was used for analyses and plotting. Simca 14.0 was used to run the PLS-DA analysis.

## 3. Results and Discussion

### 3.1. Identification of Protease-Producing Fungi

Photographs of the hydrolysis circles and their sizes of hydrolysis circles are shown in Figure 1 and Table 1. The hydrolysis circle diameters of D4, D5, D7, D11, D12, and D20 strains were larger at 1.90, 1.63, 1.57, 1.63 and 1.77 cm, respectively. The diameter of the hydrolysis circle directly corresponds to the strain’s protease production ability. Thus, D4, D5, D7, D11, D12, and D20 exhibit strong protease production ability.

#### 3.1.1. Determination of Protease Hydrolysis Activity

A standard curve was drawn with the L-tyrosine concentration as the ordinate and the absorbance value as the abscissa. According to the linear regression equation obtained from the standard curve, y = 0.0105x with a correlation (R^2^) = 0.9995. Proteases and peptidases contained in microorganisms can decompose proteins in meat products and produce polypeptides and amino acids, thus promoting the nutritional value and unique flavor of meat [27,28]. The protease activity of 10 fungi strains, obtained by preliminary screening, is shown in Table 2. D4 exhibited the highest enzyme activity (53.33 U/mg), followed by D5, D20, and D11 (43.05, 39.24, and 27.43 U/mg, respectively). The re-screening results were consistent with the preliminary screening results. Ju et al. [29] determined the protein hydrolysis activity of *Staphylococcus cohnii*, *Staphylococcus saprophyticus*, and *Staphylococcus* equorum, indicating that these three staphylococcal strains have excellent proteolytic activity, and thus they were used for subsequent experiments.

#### 3.1.2. Strain Identification

Based on the fermentation characteristics of the strains, four high-yield protease-producing strains were selected for ITS DNA sequence identification and compared with known strains from the GenBank database. A phylogenetic tree of the four fungi was constructed using MEGA 6.0. As shown in Figure 2, strains D4 and D11 were identified as *Yarrowia hollandica*, strain D5 as *Penicillium oxalicum*, and strain D20 as *Meesziomyces ophidis*. Lee et al. [30] screened *Penicillium candidum* and *Penicillium nalgiovense* and reapplied them to dry-aged beef.

### 3.2. SDS-PAGE

The changes in the sarcoplasmic protein electrophoresis pattern are shown in Figure 3. There were no significant changes in the protein bands of the control group on days 0 and 4, and the difference was approximately 17 kDa. However, during the process of sarcoplasmic protein extraction, most endogenous enzymes in the muscle are inactivated, which has little effect on the fermentation results [31]. Based on their solubility, meat proteins can be divided into three categories: myofibrillar, sarcoplasmic, and matrix proteins [32]. Sarcoplasmic proteins are naturally occurring water-soluble proteins [33], which mainly contribute to the senses [34].

The protein bands of the five treatment groups changed by varying degrees, mainly in the area around 20–135 kDa, which was significantly reduced or disappeared, and a large number of small molecular bands appeared in the area below 20 KDa, indicating that fermentation by the inoculated strain promoted the hydrolysis of sarcoplasmic proteins. The pH of the fermentation broth decreased, and the muscle tissue protease was fully activated. A large number of protein bands were degraded, and new bands were observed. The degradation of meat proteins is closely related to microorganisms, which further verifies the view that protein hydrolysis can be attributed to the combined action of endogenous cathepsins in muscles and exogenous proteases in microorganisms [35]. Careri et al. [36] observed that the sarcoplasmic protein bands of strains obtained from beef were significantly degraded at 20–48 kDa. Figure 3 shows that the 48–63 kDa protein bands in the D5 inoculation group were more visible than those in the other groups, indicating that *Penicillium oxalicum* had the strongest protein degradation capacity. Ju et al. [29] found that the protein band from the fermentation broth inoculated only with *Staphylococcus* showed different degradation effects from those of the other two groups at 26 kDa, indicating that the bacterium had a strong ability to degrade sarcoplasmic proteins when endogenous enzymes were excluded.

### 3.3. Free Amino Acid Analysis

Free amino acids play an important role in the flavor formation of fermented meat products because they are flavor compound precursors [37]. To verify the results of protein electrophoresis, we evaluated the free amino acid content of the culture broth before and after inoculation with sarcoplasmic proteins. The increase in total free amino acids may be related to the proteolytic activity of *Yarrowia hollandica* (D4 and D11), *Meesziomyces ophidis* (D20), and *Penicillium oxalicum* (D5). The total free amino acid content in the inoculated group was higher than that in the control group (225.7 mg/100 mL) (*p* < 0.05; Table 3). The D5 inoculation group showed the largest change in free amino acids, which increased to 453.1 mg/100 mL after 4 d of fermentation, followed by the D20 and D11 fermentation groups. Cathepsins in meat are likely responsible for the early hydrolysis of proteins into polypeptides, which are then hydrolyzed by microbial peptidases into small molecular free amino acids during secondary proteolysis [38]. The amount of free amino acids in the samples inoculated with single strains was considerably higher than that in the samples inoculated with mixed strains, which may be due to antagonism between the microorganisms. Competition between microorganisms for nutrients leads to a decrease in the number of microorganisms, ability to degrade proteins, and amino acid content [39].

At the end of the fermentation process, the predominant free amino acids were aspartic acid (Asp) and glycine (Gly). The concentration of aspartic acid in the fermentation broth inoculated with the mixed strains was the highest (*p* < 0.05). These amino acids affect the taste of meat products. Aspartic acid is related to umami, whereas glycine is related to sweetness [40]. The content of the branched-chain amino acids leucine (Leu) and isoleucine (Ile) in the fermentation group was higher than that in the control group (*p* < 0.05). Branched-chain amino acids can be converted into corresponding α-keto acids through microbial transaminases [41] and further metabolized into aldehydes, alcohols, and acids. For example, 3-methylbutyric acid and 2-methylbutyric acid are derived from Leu and Ile, respectively [42]. Tyrosine (Tyr) also increased significantly, providing a rich material basis for the formation of system-related volatile flavor components [36]. In summary, *Yarrowia hollandica* (D4 and D11), *Meesziomyces ophidis* (D20), and *Penicillium oxalicum* (D5) promote the hydrolysis of beef sarcoplasmic proteins and produces substances such as amino acids.

### 3.4. Changes in Volatile Flavor Components

The volatile component content following microbial fermentation is shown in Table 4. The volatile flavor components in the five inoculation groups were greater than those in the control group 4 d post-fermentation. Following fermentation, 26 volatile flavor components were detected in the D5 and F groups, 25 in the D20 group, and 18 in the blank fermentation group. Alkanes and aldehydes were the main components before and after fermentation in all groups. From Table 4, it can be seen that the total amount of volatile flavor components in the inoculated group was higher than that in the uninoculated group by 18.49% (*p* < 0.05). Among them, the total amount of volatile flavor components in the D5 fermentation group was the highest (45.47%), followed by the D11 fermentation group (43.94%), which may be related to the higher free amino acid content in the D5 group. According to previous studies [43,44], free amino acids can generate flavor compounds such as aldehydes, alcohols, and other compounds that contribute to the flavor of fermented meat products. Liu et al. [45] found that *Penicillium oxalicum* produces more than 20 novel compounds, including terpenes, diterpenes, flavonoids, and citrinin analogs, some of which exhibit anti-inflammatory and antibacterial properties.

Alcohols are mainly derived from the metabolism of carbohydrates, oxidation of fats, and reduction in fatty acid derivatization or carbonyl compounds, and the aroma of alcohols increases with the growth of the carbon chain [46]. Most alcohols have a pleasant aroma, but their threshold is high; therefore, they contribute less to the flavor of the fermentation broth [47]. The main alcohols produced in the fermentation of sarcoplasmic protein were 1-Pentanol, 1-Heptanol, 1-Octen-3-ol, Isooctanol, 1-Octanol, 1-Nonanol, and 1-Dodecanol. 1-Octene-3-ol and pentanol have a mushroom and grease flavor, respectively [48,49]. Only 1-Octene-3-ol was detected in D5, and it was either not detected or was below the detection limit in the other groups.

Aldehydes are produced through the pentose phosphate pathway and amino acid catabolism [50]. Only small amounts of octanal and non-anal compounds were detected in the control group, and no other aldehydes were detected. More aldehydes were detected in the fermentation group, and the nonanal content was relatively high. Most aldehydes are products of lipid metabolism, and the detection of aldehydes may have been caused by residual lipids during the extraction of sarcoplasmic proteins [51].

Five ketones were detected: 2,3-Butanedione, 4-Methyl-2-heptanone, Cyclohexanone, 4,6-Dimethyl-2-heptanone, and 6-Methyl-5-heptene-2-one. Diketones are products of the initial stage of the Maillard reaction with meaty and buttery aromas. Many microorganisms convert sugars into acetoin during glycolysis to avoid excessive acidification [52]. Acetoin is reversibly converted to 2,3-butanediol by butanediol dehydrogenase or to 2,3-butanedione by diacetyl reductase [53].

Alkanes are the precursors of ketones and aldehydes, which have a high threshold and potentially contribute to flavor [54]. A total of 17 alkanes were detected. Although there are many types of alkanes, they are present in relatively low amounts. The two alkanes, tetradecane and hexadecane, were present in the highest amounts.

### 3.5. Metabolic Pathway Analysis of Penicillium oxalicum (D5) Fermentation Broth

A PLS-DA model variable projection importance greater than one was used as the standard to find significant differentially expressed metabolites between the blank fermentation group and the *Penicillium oxalicum* (D5) fermentation group. The results are shown in Figure 4 and Figure 5. PLS R^2^x(cum) = 0.973, R^2^y(cum) = 1, and Q^2^(cum) = 1, indicating that the model has good predictive ability [55]. Figure 4 shows that more substances were concentrated near the D5 group, indicating that the addition of *Penicillium oxalicum* fermentation sarcoplasmic proteins could produce richer flavor substances. Figure 5 shows the differential metabolite screening of *Penicillium oxalicum* (D5) fermentation broth. 1-octen-3-ol, isoleucine, alanine, leucine, aspartic acid, tyrosine, and nonanal were identified, and the KEGG database was used to analyze the metabolic pathways; metabolic pathway bubble diagrams were obtained (Figure 6) as well as histograms of enrichment analysis (Figure 7). Wang et al. [56] found that 1-octen-3-ol, octanal, (E)-2-octenal, 1-nonenal, and (Z)-hept-2-ena were common differential flavor compounds found in inoculated sausages by PCA and OPLS-DA.

In Figure 4, according to the principle of pathway impact > 0.1 and *p* < 0.05 (−log(*p*) > 2), it was determined that the two metabolic pathways of alanine, aspartate, and glutamate metabolism and phenylalanine, tyrosine, and tryptophan biosynthesis were closely related to the formation of flavor substances in the D5 fermentation group.

In addition, the enrichment analysis showed that the differential metabolites were mainly enriched (Enrichment Ratio > 10) in Aminoacyl-tRNA biosynthesis, valine, leucine, and isoleucine biosynthesis and phenylalanine, tyrosine, and tryptophan biosynthesis (Figure 7). Among them, the metabolic pathway of phenylalanine, tyrosine, and tryptophan biosynthesis is also enriched (Figure 6). The levels of phenylalanine and tyrosine in the D5 group were higher than those in the control group (*p* < 0.05; Table 3). Combined with amino acid changes, it can be considered that phenylalanine, tyrosine, and tryptophan biosynthesis is the most relevant metabolic pathway during sarcoplasmic protein fermentation. For the low-sodium formulation treatment in ham, the metabolic pathways of palmitic acid, stearic acid, linoleic acid, and myristic acid were mainly affected, which in turn affected the formation of volatile flavor compounds [57].

## 4. Conclusions

Protease-producing microorganisms were isolated from the dry-aged beef samples. Four fungal strains were identified in this study. Strains D4 and D11 were identified as *Yarrowia hollandica*, strain D5 as *Penicillium oxalicum*, and strain D20 as *Meesziomyces ophidis*. Among them, *Yarrowia hollandica* (D4) and *Penicillium oxalicum* (D5) exhibited higher enzyme production activities. They were confirmed to have excellent ability to degrade sarcoplasmic protein, among which *Penicillium oxalicum* (D5) possessed the highest activity. Metabolic pathway analysis of the fermentation broth showed that phenylalanine, tyrosine, and tryptophan biosynthesis were the metabolic pathways most significantly related to the sarcoplasmic protein fermentation by *Penicillium oxalicum* (D5). The *Penicillium oxalicum* (D5) screened in this experiment can be subsequently applied to dry-aged beef to verify its practical effect and determine the effect of *Penicillium oxalicum* on proteins and volatile flavor components during fermentation of dry-aged beef. Overall, these study findings provide a theoretical basis for the development of starter cultures and novel dry-aging processes, potentially enhancing the development of fermented meat products.

## Figures and Tables

**Figure 1 foods-13-01038-f001:**
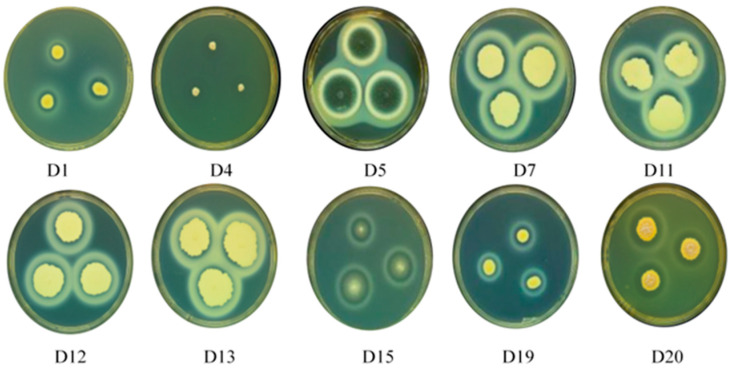
Representative image of plate cultures depicting the milk hydrolysis capabilities of several fungi (D1, D4, D5, D7, D11, D12, D13, D15, D19, and D20) isolated from dry-aged beef.

**Figure 2 foods-13-01038-f002:**
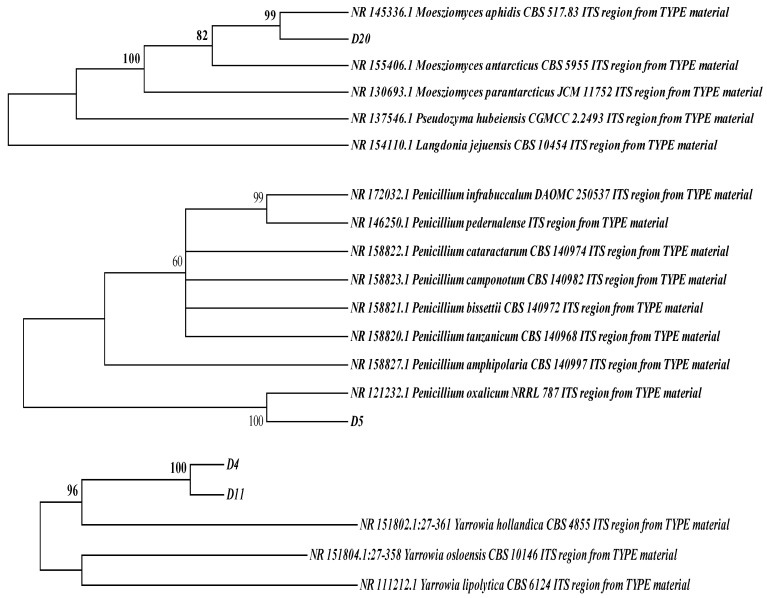
Phylogenetic tree of strains D4, D5, D11, and D20 based on ITS sequences.

**Figure 3 foods-13-01038-f003:**
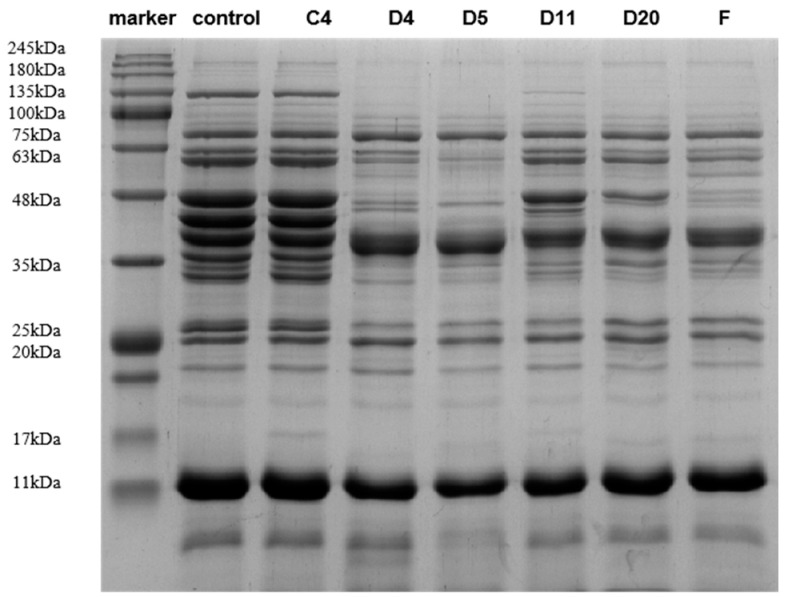
SDS-PAGE map of sarcoplasmic proteins of different strains 4 days post-inoculation.

**Figure 4 foods-13-01038-f004:**
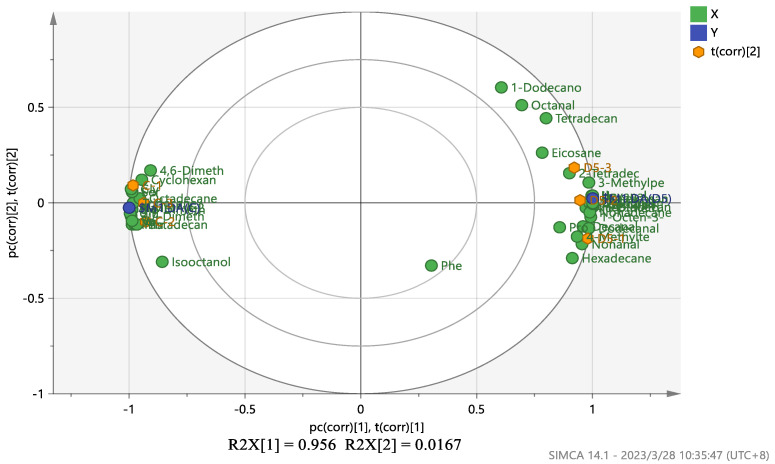
PLS-DA analysis of metabolites between blank fermentation group and *Penicillium oxalicum* (D5) fermentation group.

**Figure 5 foods-13-01038-f005:**
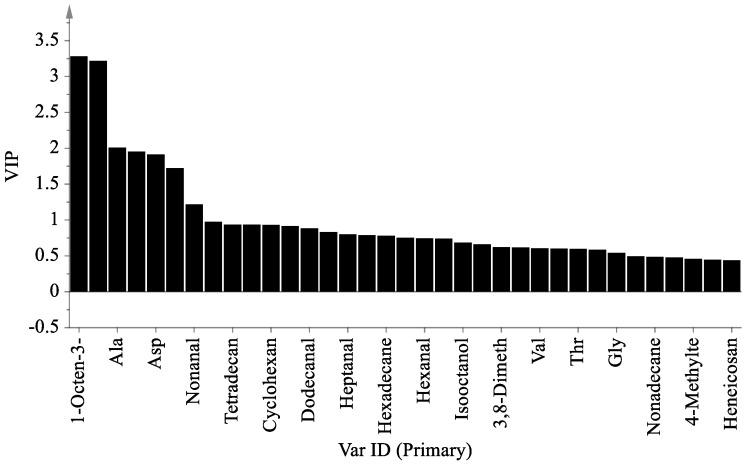
Analysis of VIP value of metabolites from fermentation broth of *Penicillium oxalicum* (D5).

**Figure 6 foods-13-01038-f006:**
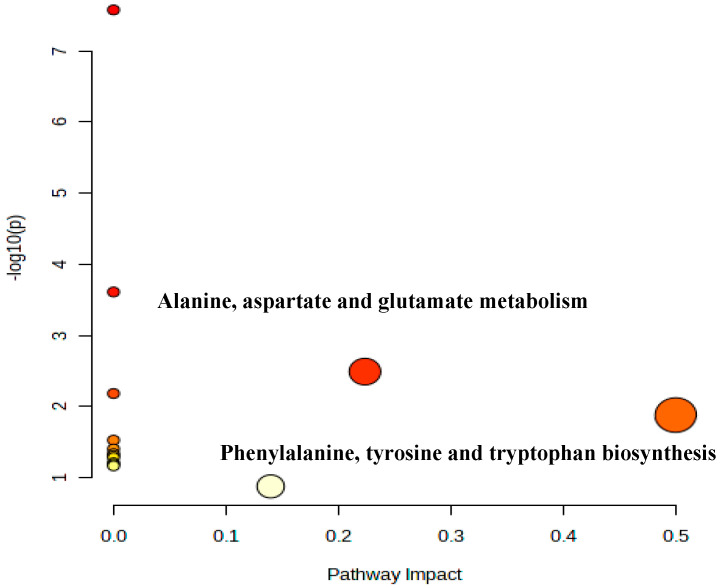
Metabolic pathway bubble diagram of *Penicillium oxalicum* (D5) from Metabo analyst.

**Figure 7 foods-13-01038-f007:**
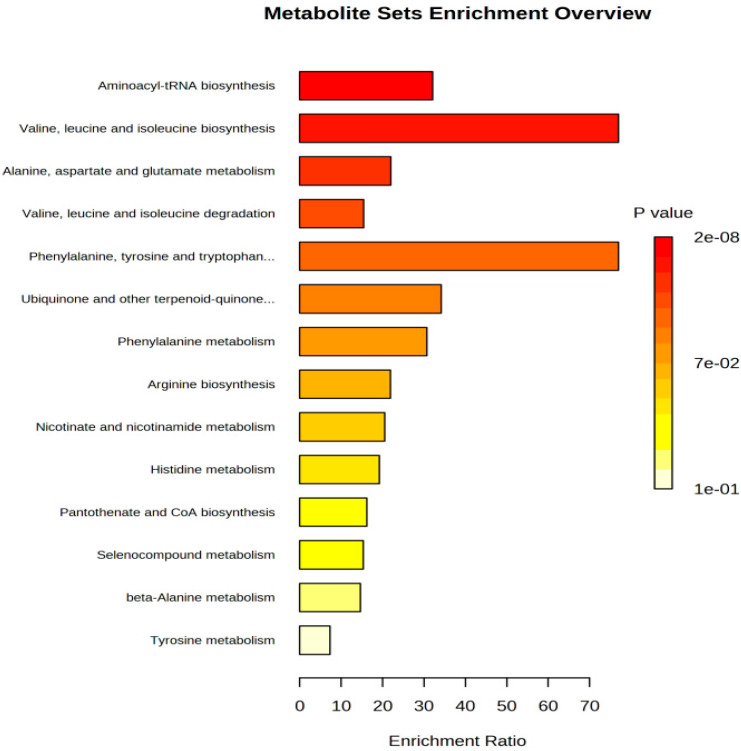
Enrichment analysis histogram of *Penicillium oxalicum* (D5) from Metabo analyst.

**Table 1 foods-13-01038-t001:** The diameter of the protease hydrolysis circles produced by each dry-aged beef isolated strain.

Strain Number	Strain Diameter (cm)	Hydrolysis Circle Diameter (cm)	Strain Diameter and Diameter of Hydrolysis Circle (cm)
D1	1.00 ± 0.10 ^c^	0.47 ± 0.12 ^e^	1.47 ± 0.06 ^c^
D4	0.57 ± 0.21 ^d^	1.90 ± 0.20 ^a^	2.47 ± 0.25 ^b^
D5	2.57 ± 0.21 ^a^	1.63 ± 0.22 ^bc^	4.20 ± 0.36 ^a^
D7	2.40 ± 0.10 ^a^	1.57 ± 0.06 ^bc^	3.97 ± 0.12 ^a^
D11	2.27 ± 0.31 ^a^	1.63 ± 0.23 ^bc^	3.90 ± 0.20 ^a^
D12	2.30 ± 0.20 ^a^	1.77 ± 0.21 ^b^	4.07 ± 0.15 ^a^
D13	2.47 ± 0.15 ^a^	1.63 ± 0.21 ^bc^	4.10 ± 0.20 ^a^
D15	1.40 ± 0.10 ^b^	0.80 ± 0.17 ^d^	2.20 ± 0.26 ^b^
D19	1.10 ± 0.17 ^bc^	1.23 ± 0.45 ^c^	2.33 ± 0.47 ^b^
D20	1.07 ± 0.21 ^bc^	1.40 ± 0.40 ^bc^	2.47 ± 0.21 ^b^

^a–e^ Means within the same column with different superscript letters indicate differences (*p* < 0.05).

**Table 2 foods-13-01038-t002:** Protease production activity of the strain.

Strain Number	D1	D4	D5	D7	D11	D12	D13	D15	D19	D20
Enzyme Activity U/mg	14.73 ± 3.43 ^f^	53.33 ± 0.76 ^a^	43.05 ± 3.43 ^b^	20.19 ± 3.43 ^de^	27.43 ± 1.52 ^c^	20.95 ± 3.43 ^de^	21.71 ± 1.14 ^d^	6.10 ± 3.05 ^g^	16.76 ± 2.29 ^ef^	39.24 ± 0.38 ^b^

^a–g^ Means within the same row with different superscript letters indicate differences (*p* < 0.05).

**Table 3 foods-13-01038-t003:** Free amino acid content 4 days post-fermentation (mg/100 mL).

FAA (mg/100 mL)	Control *	D4	D5	D11	D20	F
Asp	42.2 ± 1.7 ^f^	80.5 ± 0.4 ^d^	99.0 ± 1.7 ^c^	65.3 ± 0.1 ^e^	112.3 ± 1.4 ^b^	115.7 ± 1.2 ^a^
Glu	18.3 ± 0.9 ^a^	9.5 ± 0.3 ^bc^	5.3 ± 0.2 ^d^	4.4 ± 0.2 ^e^	10.0 ± 0.1 ^d^	8.7 ± 0.1 ^c^
Ser	18.1 ± 0.3 ^b^	14.8 ± 0.6 ^cd^	16.1 ± 0.1 ^c^	29.2 ± 1.9 ^a^	13.6 ± 0.7 ^d^	7.9 ± 0.2 ^e^
Gly	23.2 ± 0.2 ^c^	13.7 ± 0.1 ^e^	18.6 ± 0.6 ^d^	25.3 ± 2.0 ^b^	29.7 ± 0.4 ^a^	20.5 ± 0.5 ^d^
His	8.3 ± 0.6 ^b^	7.1 ± 0.5 ^b^	5.2 ± 0.1 ^c^	18.9 ± 0.6 ^a^	4.9 ± 0.6 ^c^	4.9 ± 0.3 ^c^
Thr	10.0 ± 1.2 ^a^	4.1 ± 0.0 ^c^	4.3 ± 0.2 ^bc^	5.2 ± 0.2 ^b^	2.8 ± 0.3 ^d^	2.9 ± 0.1 ^d^
Ala	65.3 ± 0.1 ^a^	3.2 ± 0.2 ^d^	2.7 ± 0.2 ^d^	9.3 ± 1.0 ^b^	5.7 ± 0.1 ^c^	5.7 ± 0.0 ^c^
Pro	6.1 ± 0.5 ^b^	4.6 ± 0.1 ^cd^	7.0 ± 0.2 ^b^	12.2 ± 1.7 ^a^	4.1 ± 0.3 ^d^	5.9 ± 0.1 ^bc^
Tyr	4.2 ± 0.1 ^e^	7.8 ± 0.2 ^d^	50.3 ± 0.2 ^a^	10.2 ± 0.1 ^c^	23.1 ± 0.1 ^b^	4.2 ± 0.1 ^e^
Val	10.6 ± 0.7 ^a^	3.5 ± 0.2 ^e^	4.8 ± 0.1 ^d^	5.7 ± 0.2 ^b c^	5.2 ± 0.2 ^cd^	6.0 ± 0.5 ^b^
Ile	5.7 ± 0.3 ^e^	96.7 ± 2.5 ^c^	166.5 ± 3.5 ^a^	100.0 ± 9.2 ^c^	132.5 ± 1.6 ^b^	52.5 ± 1.0 ^d^
Leu	8.0 ± 0.7 ^f^	41.9 ± 0.1 ^d^	67.2 ± 1.8 ^a^	44.7 ± 0.1 ^c^	53.8 ± 0.7 ^b^	25.1 ± 1.2 ^e^
Phe	6.0 ± 0.7 ^b^	4.7 ± 0.0 ^c^	6.3 ± 0.2 ^b^	8.4 ± 0.5 ^a^	4.7 ± 0.0 ^c^	4.8 ± 0.1 ^c^
Total	225.7 ± 0.7 ^f^	291.8 ± 2.2 ^d^	453.1 ± 8.8 ^a^	338.7 ± 2.1 ^c^	402.0 ± 0.3 ^b^	264.6 ± 5.3 ^e^

* Control: non-inoculated control; D4 and D11: *Yarrowia hollandica*; D20: *Meesziomyces ophidis*; D5: *Penicillium oxalicum;* F: a mixed strains (D4:D5:D11:D20) in a 1:1:1:1 ratio. ^a–f^ Means within the same row with different superscript letters indicate differences (*p* < 0.05).

**Table 4 foods-13-01038-t004:** Volatile compounds in sarcoplasmic proteins inoculated with different strains after incubation at 37 °C for 4 days (%).

Component	Control *	D4	D5	D11	D20	F
Aldehydes						
Hexanal	N.D.	N.D.	0.86 ± 0.02 ^b^	N.D.	1.35 ± 0.18 ^a^	0.36 ± 0.07 ^c^
Heptanal	N.D.	N.D.	0.99 ± 0.03 ^b^	N.D.	1.64 ± 0.72 ^a^	N.D.
Octanal	0.70 ± 0.03 ^b^	0.91 ± 0.02 ^ab^	1.05 ± 0.31 ^a^	0.86 ± 0.16 ^ab^	1.01 ± 0.13 ^ab^	1.14 ± 0.20 ^a^
Nonanal	1.82 ± 0.66 ^d^	6.96 ± 0.24 ^b^	4.24 ± 0.27 ^c^	1.79 ± 0.88 ^d^	11.63 ± 1.82 ^a^	11.74 ± 1.08 ^a^
Decanal	N.D.	0.28 ± 0.10 ^c^	1.54 ± 0.40 ^b^	9.20 ± 1.34 ^a^	1.65 ± 0.29 ^b^	2.04 ± 0.60 ^b^
Undecanal	N.D.	N.D.	N.D.	N.D.	0.49 ± 0.07 ^b^	0.68 ± 0.06 ^a^
Dodecanal	N.D.	0.96 ± 0.38 ^b^	1.23 ± 0.21 ^b^	0.25 ± 0.01 ^a^	1.09 ± 0.12 ^b^	1.33 ± 0.13 ^b^
Hexadecanal	N.D.	N.D.	N.D.	0.83 ± 0.05 ^a^	0.82 ± 0.03 ^a^	N.D.
Alcohols						
1-Pentanol	N.D.	0.17 ± 0.02 ^b^	N.D.	0.68 ± 0.49 ^a^	N.D.	N.D.
1-Heptanol	N.D.	N.D.	N.D.	0.65 ± 0.21 ^a^	0.43 ± 0.03 ^b^	0.47 ± 0.11 ^b^
1-Octen-3-ol	N.D.	N.D.	16.86 ± 1.82	N.D.	N.D.	N.D.
Isooctanol	1.51 ± 0.39 ^a^	1.29 ± 0.18 ^ab^	0.68 ± 0.16 ^cd^	0.54 ± 0.26 ^d^	1.04 ± 0.08 ^bc^	0.83 ± 0.03 ^cd^
1-Octanol	N.D.	N.D.	0.98 ± 0.13 ^a^	1.01 ± 0.09 ^a^	0.59 ± 0.10 ^b^	0.95 ± 0.18 ^a^
1-Nonanol	N.D.	1.60 ± 0.05 ^a^	N.D.	1.05 ± 0.01 ^b^	0.53 ± 0.15 ^c^	1.79 ± 0.42 ^a^
1-Dodecanol	1.07 ± 0.40 ^b^	N.D.	1.42 ± 0.38 ^b^	2.02 ± 0.04 ^a^	0.46 ± 0.06 ^c^	N.D.
2-Tetradecanol	N.D.	N.D.	0.98 ± 0.40	N.D.	N.D.	N.D.
Ketones						
2,3-Butanedione	N.D.	N.D.	N.D.	N.D.	N.D.	0.38 ± 0.01
4-Methyl-2-heptanone	1.67 ± 0.12 ^a^	0.60 ± 0.36 ^b^	0.59 ± 0.06 ^b^	1.31 ± 0.31 ^a^	0.85 ± 0.30 ^b^	0.55 ± 0.24 ^b^
Cyclohexanone	3.72 ± 0.24 ^b^	2.45 ± 0.60 ^c^	2.29 ± 0.36 ^c^	6.20 ± 0.73 ^a^	3.34 ± 0.34 ^b^	2.89 ± 0.08 ^bc^
4,6-Dimethyl-2-heptanone	0.94 ± 0.39 ^a^	N.D.	N.D.	0.18 ± 0.01 ^c^	0.78 ± 0.03 ^ab^	0.56 ± 0.15 ^b^
6-Methyl-5-heptene-2-one	N.D.	N.D.	N.D.	N.D.	0.47 ± 0.02 ^a^	0.38 ± 0.09 ^b^
Alkanes						
2,4,4-Trimethylhexane	N.D.	N.D.	N.D.	N.D.	0.18 ± 0.03 ^a^	0.15 ± 0.07 ^a^
3,7-Dimethylnonane	N.D.	N.D.	N.D.	N.D.	0.14 ± 0.01 ^b^	0.21 ± 0.02 ^a^
3,7-Dimethyldecane	0.54 ± 0.05 ^b^	0.30 ± 0.05 ^c^	N.D.	1.31 ± 0.06 ^a^	0.34 ± 0.11 ^c^	N.D.
3,8-Dimethylundecane	0.95 ± 0.06 ^a^	0.12 ± 0.01 ^c^	0.34 ± 0.07 ^bc^	1.21 ± 0.44 ^a^	0.51 ± 0.04 ^b^	0.25 ± 0.12 ^bc^
4,6-Dimethyldodecane	0.23 ± 0.08 ^c^	0.51 ± 0.06 ^b^	0.92 ± 0.13 ^a^	0.05 ± 0.02 ^d^	0.47 ± 0.07 ^b^	0.63 ± 0.17 ^b^
Tridecane	0.57 ± 0.09 ^a^	0.58 ± 0.24 ^a^	N.D.	N.D.	0.46 ± 0.05 ^b^	0.43 ± 0.05 ^b^
Tetradecane	0.92 ± 0.16 ^b^	1.53 ± 0.35 ^b^	2.59 ± 1.04 ^a^	N.D.	2.28 ± 0.17 ^a^	2.44 ± 0.99 ^a^
2-Methyl-n-tridecane	0.19 ± 0.03 ^b^	N.D.	0.42 ± 0.02 ^a^	N.D.	N.D.	N.D.
4-Methyltetradecane	0.37 ± 0.12 ^c^	N.D.	0.72 ± 0.02 ^b^	0.86 ± 0.02 ^a^	N.D.	N.D.
Pentadecane	0.88 ± 0.06 ^a^	N.D.	0.50 ± 0.04 ^c^	0.91 ± 0.01 ^a^	N.D.	0.71 ± 0.14 ^b^
3-Methylpentadecane	N.D.	0.16 ± 0.04 ^b^	0.60 ± 0.09 ^a^	N.D.	N.D.	N.D.
Hexadecane	1.27 ± 0.40 ^c^	1.53 ± 0.19 ^c^	2.31 ± 0.17 ^b^	6.06 ± 0.80 ^a^	1.49 ± 0.15 ^c^	1.17 ± 0.09 ^c^
Heptadecane	N.D.	N.D.	1.36 ± 0.06	N.D.	N.D.	N.D.
Octadecane	0.83 ± 0.06 ^a^	0.13 ± 0.02 ^c^	0.56 ± 0.05 ^b^	0.72 ± 0.21 ^ab^	N.D.	N.D.
Nonadecane	N.D.	N.D.	0.37 ± 0.05 ^a^	N.D.	N.D.	0.30 ± 0.08 ^b^
Eicosane	0.33 ± 0.05 ^b^	1.56 ± 0.03 ^a^	0.78 ± 0.30 ^b^	1.54 ± 0.35 ^a^	N.D.	1.32 ± 0.48 ^ab^
Heneicosane	N.D.	N.D.	0.30 ± 0.01	N.D.	N.D.	N.D.
Others						
Phenylethyl Alcohol	N.D.	N.D.	N.D.	4.72 ± 1.05 ^a^	N.D.	0.32 ± 0.04 ^b^
Total	18.49 ± 1.60 ^c^	21.62 ± 0.72 ^c^	45.47 ± 0.63 ^a^	43.94 ± 0.87 ^a^	34.01 ± 2.51 ^b^	33.80 ± 3.06 ^b^

* Control: non-inoculated control; D4 and D11: *Yarrowia hollandica*; D20: *Meesziomyces ophidis*; D5: *Penicillium oxalicum*; F: a mixed strains (D4: D5: D11: D20) in a 1:1:1:1 ratio. ^a–d^ Means within the same row with different superscript letters indicate differences (*p* < 0.05). N.D.: not detected.

## Data Availability

The original contributions presented in the study are included in the article, further inquiries can be directed to the corresponding authors.

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
