# Peer review of "Hydrolysis of Beef Sarcoplasmic Protein by Dry-Aged Beef-Isolated Penicillium oxalicum and Its Associated Metabolic Pathways"

_foods, 2024, doi:10.3390/foods13071038_

Round 1
Reviewer 1 Report
Comments and Suggestions for Authors
There is much information stated as fact without citing the source of the information. The missing information in the materials and methods prevent determination of the appropriateness of the procedures and duplication of the experimental conditions by other scientists. The discussions of results were missing or minimal in some sections.
Line(s) Comment
14 cattle is plural; “Yanbian cattle have a”
15 Topsides do not have low consumer preference in many countries. Delete this from the sentence since foods is an international scientific journal. It is more appropriate to give the scientific muscle name rather than the trade name “topside.”
18 “aged and then screened”
22 Implied comparison; “was stronger than the other two fungi.”
24-25 It is unclear why quotation marks are in the sentence.
32-34 Reference(s) are needed to substantiate that the information is factual.
38 “But weight and trimming losses from long-term dry”
39-42 Reference(s) are needed to substantiate that the information is factual.
44-47 Reference(s) are needed to substantiate that the information is factual.
51-55 Reference(s) are needed to substantiate that the information is factual.
66-67 The number of cattle and the number of topside pieces should be given.
77 The method and equipment for homogenization should be described.
81 The method of determining proteolytic cycle activity should be described.
107 The materials and method for filtering supernatant must be given.
117-118 The conditions and time for gel electrophoresis must be given.
125 The model and manufacturer of the gel analyzer must be given.
134 The model and manufacturer of the amino acid analyzer must be given.
137 The method to extract volatile flavor substances by SPME must be described.
140 The type and parameters of the sand bath must be given.
140 The model and manufacturer of the gas chromatograph must be given.
152 One-way ANOVA is not appropriate unless the number of experimental replications are given and it is shown that the replication effect or replication x treatment effect are not significant.
158-217 There was no discussion of the results, such as comparison of findings from other similar studies.
159-164 This is a description of the isolation of fungi with proteolytic ability and should be in section 2.2.1 of materials and methods.
173 An explanation should be provided for the lack of statistical analysis of data in Table 1. The strain numbers should be described in a footnote so the table can be interpreted independently from the text.
185 An explanation should be provided for the lack of statistical analysis of data in Table 1. The strain numbers should be described in a footnote.
210-217 The order is not logical. L. 214-217, then L. 212-214, then L. 210-214.
247-248 Reference(s) are needed to substantiate that the information is factual.
253-254 Use of “significant” (or “significantly”) and the probability level here and in subsequent use is redundant.
256-258 It is not generally considered that calpain enzymes degrade sarcoplasmic proteins.
262-263 Reference(s) are needed to substantiate that the information is factual.
265 The strain numbers should be described in a footnote.
276-277 Reference(s) are needed to substantiate that the information is factual.
280 “the potential to alter the flavor” because there was no evidence presented that umami, sweetness, or any of the aldehydes, alcohols, or acids improve flavor.
300-302 Reference(s) are needed to substantiate that the information is factual.
304 Reference(s) are needed to substantiate that the information is factual.
310-311 Reference(s) are needed to substantiate that the information is factual.
314-316 Reference(s) are needed to substantiate that the information is factual.
319-320 Reference(s) are needed to substantiate that the information is factual.
325 The strain numbers should be described in a footnote.
327-339 There was no discussion of these results with results of other studies.
328-330 This methodology must be described in section 2.6 of the materials and methods.
331-332 The basis for stating the model has high predictive ability must be stated.
337-338 These methods (KEGG database analyses of the metabolic pathways, pathway bubble diagrams, histograms) must be described in section 2.1 of the materials and methods.
349 It should be indicated that this is analysis is of differentially expressed metabolites between the blank fermentation group and the Penicillium oxalicum (D5) fermentation group.
351 It should be indicated that this analysis is of metabolite screening of Penicillium oxalicum (D5) fermentation broth.
352-364 There was no discussion of these results with results of other studies.
366 It should be indicated that this analysis is of metabolite screening of Penicillium oxalicum (D5) fermentation broth.
381 It should be indicated that this analysis is of metabolite screening of Penicillium oxalicum (D5) fermentation broth.
383-398 This is a summary of results rather than explaining the novelty, usefulness, and/or importance of the results.
433 Journal title is not abbreviated as in the other citations.
Comments on the Quality of English Language
Generally language was easy to understand.
Author Response
Thanks for reviewer’s comments. We have uploaded into the system the details of the changes to the point by point description of the manuscript and the responses to the referees’ comments.

Reviewer 2 Report
Comments and Suggestions for Authors
Dear Authors,
this manuscript is interesant and good written,
I have some question or suggestion:
Line 69 there is Dry ageing of the obtained topside pieces was carried out for 40 d as previously described [11]. and I can't find these article and You didn't give the DOI - add a DOI, or add the desripcion about dry ageing;
Line 70 "Three samples were used as blank controls" and what about this samples? another "were collected at 10-, 20-, 30-, and 40 d of dry aging and then packaged and stored at -80 ℃ until further use" and the control samples?
Line 134 "automatic amino acid analyzer" - please give the name of this amino acid analyzer
Line 113 "Inoculation of 2% of single strains of Yarrowia hollandica D4 and D11" - what was the differences between D4 and D11?
Line 168 what is D7, D12, D13, D15, D19?
Table 1. please add the statistical significances; in columns and in rows
Table 2. please add the statistical significances; in rows;
Discussion is conspicuously missing from the manuscript. The work is good, but without good discussion. I understand that there is little in the literature, but there are 4-6 items that the authors will definitely find to compare with their own results. PLEASE ADD.
Good luck!
Comments on the Quality of English Language
Minor editing of English language required
Author Response

(The authors gave the same response as above.)

Round 2
Reviewer 1 Report
Comments and Suggestions for Authors
Line(s) Comment
16 The muscle used instead of the jargon name topside should be used.
37 Delete “from”
66 The specific muscle(s) used must be specified. “Topside” is an industry term, not a scientific description of the tissue used.
196, 213, 288, 349 Use of “significant” and the probability level in the same sentence is redundant.
Author Response
Thanks for reviewer’s comments. All questions have corrected in the revised manuscript. Furthermore, the point-to-point responses have also been presented in the response letter.
